# Truths and Myths in Pediatric Migraine and Nutrition

**DOI:** 10.3390/nu13082714

**Published:** 2021-08-06

**Authors:** Laura Papetti, Romina Moavero, Michela A. N. Ferilli, Giorgia Sforza, Samuela Tarantino, Fabiana Ursitti, Claudia Ruscitto, Federico Vigevano, Massimiliano Valeriani

**Affiliations:** 1Department of Neuroscience, Bambino Gesù Children Hospital, IRCCS, 00165 Rome, Italy; romina.moavero@opbg.net (R.M.); michela.ferilli@opbg.net (M.A.N.F.); giorgia.sforza@opbg.net (G.S.); samuela.tarantino@opbg.net (S.T.); fabiana.ursitti@opbg.net (F.U.); federico.vigevano@opbg.net (F.V.); massimiliano.valeriani@opbg.net (M.V.); 2Child Neurology Unit, Systems Medicine Department, Tor Vergata University Hospital of Rome, 00165 Rome, Italy; claudia.ruscitto@opbg.net; 3Center for Sensory-Motor Interaction, Denmark Neurology Unit, Aalborg University, 9100 Aalborg, Denmark

**Keywords:** migraine, diet, nutrients, allergy, food, ketogenic diet

## Abstract

The link between migraine and nutrition can be explored from several points of view. Lifestyle and, in particular, aspects of nutrition can have a significant impact on the course of pediatric migraine. In addition, some dietary treatments, such as the ketogenic diet, and some active ingredients present in foods (nutraceuticals) may have a therapeutic effect on migraine. A diet that can control weight gain and obesity has beneficial effects on migraine severity. On the other hand, when we talk about the link between nutrition and headaches, it is also necessary to point out that some public information is actually fake news that has no scientific basis. The purpose of this review is to provide an update on the salient points linking pediatric migraine to nutritional principles, focusing on the relationship between weight and headaches, the therapeutic effect of food for medical purposes, the ketogenic diet as a migraine treatment, and the relationship between migraine and dietary habits.

## 1. Introduction

Migraine is a very common neurological disorder in children and adolescents. It can become very disabling due to the intensity and high recurrence of the attacks [1]. The pathogenesis of migraine is multifactorial and involves genetic, neuronal, and vascular mechanisms [2]. Among the environmental factors that may influence the course of the disease, nutrition is one of the most discussed in the scientific community [3]. There is a heated debate on how certain foods can act as favorable or protective factors in relation to migraine attacks [4,5]. An unhealthy diet seems to favor the onset of migraine and be associated with more serious phenotypes of the disease [6]. This applies, in particular, to patients who are overweight and present a pro-inflammatory state [7]. Lastly, it is very important to underline that the focus on nutrients and dietary factors can have important therapeutic implications at pediatric age [5]. In the literature, there are data on the ketogenic diet and some nutraceuticals for migraine treatment in adults and children [8,9]. The effect of nutrients on migraine also opens up perspectives regarding possible future therapeutic strategies, from changes in diet to the production of synthetic products that can be used for migraine prophylaxis [5].

In this review, we will focus on the role of nutrients in pediatric migraine, analyzing the pros and cons of a possible association between nutrients and migraine onset and/or severity. Our aim is to disentangle what is scientifically demonstrated in this field (truths) from what is often accepted uncritically, even in the absence of any scientific background (myths) (Table 1).

## 2. Foods as Trigger Factors in Pediatric Migraine

Dietary habits play an important role in precipitating headaches in children and adolescents with migraine. Several studies conducted both in children and adults showed that the percentage of migraine patients reporting a particular food or drink as a migraine attack trigger varies from 7% to as high as 44%, with many subjects reporting more than one dietary trigger [10,11]. A survey on 120 children with migraine showed dietary factors responsible for triggering an attack in 38% of cases (chocolate: 17%; cheese: 16%; citrus fruits: 5%) [5]. A more recent study by Neut et al. showed that 32.3% of children and adolescents with migraine reported food trigger factors (chocolate: 11.8%; colas: 8.8%; soft drinks: 3.9%; citrus fruits: 3.9%; cheese: 3.9%) [12].

Several foods and drinks have been thought to trigger migraine attacks: chocolate, coffee, nuts, salami, alcoholic beverages (especially red wine and beer), milk, citrus fruits, and cheese. The most frequent ingredients claimed to have a pro-migraine effect were caffeine, monosodium glutamate, artificial sweeteners, nitrites, gluten, and biogenic amines (egg, histamine, tyramine, and phenyl ethylamine) [13,14].

Dietary factors may act on migraine through different mechanisms, including their effects on neuropeptides, neuroreceptors, and ion channels, sympathetic nervous system and cerebral glucose metabolism, and/or by causing inflammation, the release of nitric oxide, vasodilation, and vasoconstriction [15,16,17]. The effects of dietary triggers on migraine patients could depend on dosage, timing of exposure, and genetic factors [18]. It is possible that an isolated trigger is insufficient to precipitate a migraine attack, and this is why migraine patients usually recognize multiple dietary triggers [19]. While there is a vast literature on the relationship between migraine and dietary triggers, recent population studies and prospective studies, with and without dietary interventions, have been published only in adult patients [20].

In conclusion, the relationship between food and migraine remains unclear and not all pro-migraine foods will trigger a headache attack in every patient with migraine [17]. Therefore, patients with migraine should not avoid all the foods that we will describe later, unless a clear association between these factors and their headache is demonstrated.

### 2.1. Chocolate

Chocolate has been self-reported to be a precipitant for headaches in 2–22% of adult people with migraine [21,22]. In a prospective observational case series aimed at evaluating the effect of the exclusion of frequently-consumed foods in a cohort of children with headaches, chocolate was found to be a trigger factor in 22% of children [23].

While many studies have suggested a link between chocolate and migraine, the underlying physiological mechanisms are unclear [24,25,26]. Chocolate contains various polyphenols, among which are flavanols. Flavanols could stimulate endothelial nitric oxide (NO) synthase activity, leading to an increase in NO generation, which is responsible for vasodilation and blood pressure reduction [27]. Chocolate also contains phenylalanine, which has vasoconstrictive properties and may initiate a headache by the alteration of the cerebral blood flow and the release of norepinephrine from the sympathetic nerve cells [17]. Another mechanism by which chocolate may trigger headaches is an increase in serotonin level [28]. In their review, Nowaczewska et al. identified 25 studies investigating the prevalence of chocolate as a trigger factor in patients with migraine. Among them, a small proportion of migraine patients reported chocolate as a trigger factor. All provocative studies have failed to confirm that chocolate can trigger migraine attacks. Based on this review of the current literature, there is insufficient evidence that chocolate can be considered as a migraine trigger [29].

### 2.2. Caffeine

Caffeine is the most widely used psychostimulant worldwide [13]. It is found in diet products, such as tea, coffee, chocolate, soft drinks, and energy drinks. Children, especially adolescents, drink large volumes of caffeine-containing drinks daily [17].

While the correlation between caffeine and migraine is widely studied in the adult population, data are lacking in children and adolescents.

Caffeine has been linked with migraine for many years, either as an effective treatment or as a trigger [29,30]. Headache patients benefit from the use of caffeinated agents, either alone or in combination with other analgesic treatments, such as aspirin and acetaminophen [31,32,33]. Derry et al., in their review, evaluated the relative efficacy of a single dose of an analgesic plus caffeine against the same dose of the analgesic alone. They concluded that about 5% to 10% more participants achieved a good level of pain relief with the addition of caffeine [34].

Caffeine may trigger headaches in two possible ways: (1) drinking coffee or other caffeinated beverages that may start a migraine attack; or (2) withdrawing from caffeine after chronic exposure. Withdrawal symptoms that include headaches can start as early as 12–24 h after abstinence from caffeine, peak at 20–51 h, and last for 2–9 days [4,30,33,35,36]. The median percentage of individuals that experience caffeine withdrawal headaches after abstinence is 47% [35]. In their review, Nowaczewska et al. included twenty-one studies that evaluated the prevalence of caffeine/caffeine withdrawal as a migraine trigger. In most of the studies, patients were asked retrospectively to recall their usual headache trigger. Caffeine/caffeine withdrawal was found to be a migraine trigger in a small percentage of participants (ranging from 2% to 30%) [32].

Since the structure of caffeine is similar to that of adenosine, the effects of caffeine on nociception are primarily attributed to its nonselective antagonism of the adenosine A1 and A2 receptors [37]. Adenosine is one of the neuromodulators that contribute to the pathophysiology of migraine, although the mechanisms remain elusive. Chronic blocking of adenosine receptors by habitually drinking coffee seems to increase the burden of migraine, and this is thought to result in an increased sensitivity to adenosine, which is evident when caffeine is withdrawn [30].

### 2.3. Alcohol

A recent report found that children and adolescents from 12 to 17 years accounted for 25% of alcohol consumed in the USA, including wine in addition to beer and spirits [38]. Given that 29–36% of migraineurs self-report that alcohol precipitates migraine attacks [39,40], alcohol should be included among the dietary factors possibly precipitating migraine, even in pediatric age. Despite the widespread use of alcohol among adolescents, there are no studies evaluating alcohol as a trigger factor for migraine in pediatric patients.

Several studies in adults reported a high percentage of patients who considered red wine to be the most frequent trigger among alcoholic drinks [41,42]. However, according to other studies, migraine attacks could be more frequently triggered by white wine or sparkling wine or spirits and beer [40,43,44]. Dueland, in his review, analyzed science reports, experimental studies, and clinical and epidemiological data on alcohol as a headache trigger. Large-scale epidemiological studies showing that the consumption of alcohol is reduced among people who report headaches or have a diagnoses of primary headache disorders [42]. This suggests that previous experiences of alcohol as a headache trigger may have reduced the alcohol consumption among headache sufferers. 

Regarding the putative mechanism of action, it is possible that alcohol itself can trigger headaches, especially when ingested in large quantities, and that some components of the alcoholic drinks can reinforce the action of alcohol [45]. There is still uncertainty regarding what the mechanism leading to alcohol-triggered headaches may be, although it is thought to be multifactorial and involve different substances, such as histamine, tyramine, sulfites, phenylethylamine, flavonoid phenols, and 5-hydroxytryptamine system [42,45,46].

### 2.4. Aspartame, Monosodium Glutamate, and Nitrites

Aspartame, an artificial sweetener added to many foods and beverages, may trigger headaches in susceptible individuals [11,47]. It is composed of phenylalanine (50%), aspartic acid (40%), and methanol (10%). These substances can alter the synthesis and release of neurotransmitters (dopamine, serotonin, and norepinephrine), which are known regulators of neurophysiological activity [48]. Aspartame acts as a chemical stressor by elevating plasma cortisol levels and causing the production of excess free radicals, which may increase the brain’s vulnerability to oxidative stress. This, in turn, may have adverse effects on neurobehavioral health [49]. All the reported studies were conducted in adult patients. The effect of aspartame in pediatric migraine needs to be confirmed by controlled studies.

It has been reported that the consumption of aspartame could cause neurological and behavioral disturbances, including headaches (45%), dizziness (39%), confusion/memory loss (29%), and insomnia (14%) [46]. Several studies exploring aspartame-triggered headaches failed to find any difference in headache frequency between individuals exposed to either a placebo or aspartame [49,50,51,52]. However, another two studies demonstrated a higher frequency of headaches in subjects exposed to aspartame than to a placebo [11,47]. In a case series, Newman and Lipton reported two persons whose migraines were triggered after the ingestion of rizatriptan melting tablets, which contain aspartame as an additive substance [53].

Monosodium glutamate (MSG) is used worldwide as a flavor enhancer; it is used in a variety of processed foods, including frozen or canned foods, soups, snack foods, salad dressing, seasoning salts, ketchup, and barbecue sauces [54]. In 1968, Kwok reported the so-called Chinese Restaurant Syndrome (CRS), which appeared after the consumption of Chinese dishes and includes some transient symptoms, such as headache, flushing, paresthesias, sweating, palpitations, and general weakness [55]. In their review, Obayashi and Nagamura reported the results of a systematic review of the available human studies of MSG, focusing on the causal relationship between MSG intake and headaches. 

The analysis was conducted by separating the human studies in which MSG was administered with and those in which MSG was administered without food due to the significant difference in the kinetics of glutamate between those conditions. Among the studies in which MSG was administered with food, none showed a significant difference in the incidence of headaches, except for the female group in one study. In all the studies in which MSG was administered without food, a significant difference in the headache incidence was found. Many of the studies also involved the administration of MSG in a solution at high concentrations. Since the distinctive MSG is readily identified at such concentrations, these studies were thought not to be properly blinded. Because of the absence of proper blinding, and the inconsistency in the findings, the authors conclude that further studies are required to evaluate the relationship between MSG ingestion and headaches [56].

Nitrites are often used as preservatives for meats, such as bacon, sausage, ham, and lunch meats [15]. In 1972, Henderson and Raskin published a case of headaches triggered by nitrites, which later became known as the “hotdog headache” [57].

There have been fewer studies on nitrites triggering headaches in recent years. A study found that 5% of persons with migraine were more likely to have an attack after consuming nitrites [58]. More recent studies suggest that NO plays an important role in nitrate-induced migraine attacks. NO acts on the vascular endothelium by producing vasodilatation, which may induce migraine attacks [38]. NO is also produced during cortical spreading depression and may be involved in pain modulation within the central trigeminal pathway [59].

## 3. Migraine and Food Allergy

Food allergy is an adverse reaction to food that is caused by an abnormal immunological reaction mediated by antibodies of the IgE class, which react towards protein-based food components [60]. Signs and symptoms of food allergy appear shortly after food intake, from a few minutes to a few hours. The foods responsible for the majority of allergic reactions are milk, eggs, peanuts, fish, nuts, and soy in children and peanuts, tree nuts, fish, shellfish, soy, vegetables, and fruit in adults [61]. The relationship between migraine and food allergy is controversial, generating public and medical interest but an equal amount of skepticism [62]. Several studies conducted on adults have tried to understand if there are any connections between food allergies and migraines and if oligoantigenic diets could have any benefit [63,64,65]. An improvement was observed in migraine patients after receiving an oligoantigenic diet, which is supposed to be due to an immunological mechanism involving both the removal of antigens and the elimination of the high amounts of amines present in these foods [64]. However, this hypothesis is not supported by experimental studies. Other authors even found conflicting data [16]. Therefore, it is not possible to define whether the improvement in migraine observed during oligoantigenic diets is due to the control of immuno-mediated mechanisms or simply to the elimination of possible trigger foods. 

In 1983, Egger et al. evaluated the effect of an oligoantigenic diet (consisting of one meat, one carbohydrate, one fruit, one vegetable, water, and vitamin supplements) in 88 children affected by migraine. Fifty-two percent of patients had positive skin-prick tests to one or more of the antigens used routinely for identifying atopic subjects (timothy grass pollen, dermatophagoide, cat fur, cows’ milk, and hens’ eggs). A second phase of the study, completed by 40 patients, consisted in the reintroduction of possible active foods. The authors specified that patients with provoked symptoms were investigated further by administering tartrazine and benzoic acid, which can promote headaches [66]. The study had several limitations. First, the criteria used for the migraine diagnosis were not standardized. Second, although the symptomatology of 78 out of 88 patients who completed the oligoantigenic diet was resolved, the authors did not specify whether the improvement concerned only migraine or other disorders, such as malaise and abdominal pain. Third, all the patients continued traditional drug therapy for migraine. Fourth, since no difference in response to diet was noted between patients with or without positive prick tests, this study does not clarify whether the observed improvement was only related to the removal of foods that can trigger migraines or to an immune-mediated mechanism. 

We can conclude that, although the role of some foods as triggers for migraine attacks is recognized, and some dietary regimes may be useful for the treatment of migraine, food allergy cannot be considered as a cause of migraine.

## 4. Migraine and Obesity

In pediatric age, overweight and obesity are defined as a body mass index (BMI) above the 85th and the 95th percentile, respectively, for children and teens of the same age and sex. Over the last decades, in developed countries, the obesity rate has increased globally in children of all ages. From 1980 to 2013, pediatric overweight and obesity rose to 23.8% in boys and 22.6% in girls [6]. Therefore, we can conclude that migraine and obesity represent common and significant health problems in pediatric age, which are often associated with high healthcare costs, a poor quality of life, and social isolation [6,7,67,68,69,70,71]. These negative effects may be magnified when both conditions overlap.

In recent years, an increasing body of literature has explored the association between migraine and obesity/overweight and the possible mechanisms behind their comorbidity, both in adulthood [72,73,74,75,76,77] and at pediatric age [6,7,67,78,79]. While many studies supported an association between overweight/obesity and headaches in children [6,7,68,78,79], some authors did not confirm this association [80,81,82,83]. The first pediatric study to demonstrate an association between obesity and primary headache was conducted in Israel by Pinhas-Hamiel et al. (2008). This study, including 273 children aged between 9 and 17, showed a higher prevalence of episodic migraine in obese children (8.9%), compared with overweight (4.4%) and normal-weight (2.5%) children [84]. A more recent cross-sectional population study showed an increased risk of headaches (40%) in overweight or obese adolescents, compared to normal-weight adolescents. In particular, the risk of having migraine has been found to be 60% higher in overweight and obese adolescents [85]. There also appears to be a higher frequency of migraines in obese female patients than in males. One study shows a higher prevalence of headache in obese girls (20.3%), but not boys, compared with normal-weight (7.7%) young females [84]. Another study showed a 5-fold higher risk for overweight females with migraine, when compared with males [78]. Previous studies have explored the role of being overweight not only on the prevalence, but also on the severity of headaches, particularly the intensity and frequency of the attacks [78,85,86,87]. Obese children showed higher headache attacks per month (5.3 ± 2.6), compared not only with normal-weight (3.6 ± 2.2), but also with overweight children (4.4 ± 2.4) [87]. Moreover, a high prevalence of chronic daily headache (CDH) was described in the overweight population [6,78,88]. Among children with CDH, the prevalence of overweight or obese children may be nearly twice (23%) that of normal-weight children (12%) [88]. More recently, we showed a high prevalence of overweight/obesity (50.4%) among children/adolescents with migraine. Moreover, 64.7% of patients with a high frequency of headache attacks were overweight, while 35.3% were normal-weight [79]. 

There is evidence that a decrease in body weight may lead to a reduction in the incidence and severity of headaches in both the adult and pediatric population [67,86,89,90]. In 2013, a multicenter study conducted on obese adolescents with migraine (14–18 years old) explored the effects of an interdisciplinary 12 month-long intervention program (including physical exercise, dietary education, and behavioral therapy) on headache outcomes. It was shown that weight reduction was significantly associated with a reduction in headache frequency and intensity [67].

Regarding the studies on the association between pediatric migraine and obesity, a retrospective study showed no significant difference in the percentage of children suffering from episodic or chronic migraine who were obese, compared to the general population with a normal weight [81]. In agreement with these data, a study involving children with migraine and tension-type headaches (TTH) showed a high prevalence of normal weight in both patients with migraine (73.5%) and TTH (68.2%). In particular, among children suffering from migraine, 19.2% were overweight, and only 7.3% were obese [80]. 

The psychological aspects of the relationship between migraine and overweight also need to be considered [78,79]. Both conditions may be associated with anxiety and depression, which, in turn, may correlate with disability [91,92,93]. Our recent data showed that, at pediatric age, anxiety symptoms might be a vulnerability factor influencing not only the frequency of migraine attacks, but also the relationship between weight and migraine severity [79].

A sedentary lifestyle is another risk factor contributing to the association between headaches and overweight. The lack of physical activity is associated with an increased risk of attacks in adults with migraine (21%) [94] and augmented risk of migraine (50%) in adolescents [85]. Moreover, obese migraineurs perform less physical activity than obese people without migraines [87].

The etiopathological link between obesity and migraine is probably multifactorial [6,7,73,78]. Changes in hypothalamic activation, bioactive neurotransmitters, and neuropeptides that modulate energy homeostasis, such as serotonin, calcitonin gene-related protein (CGRP), orexin, leptin, and adiponectin, play a role in both migraine and nutrition [7,73]. The nucleus of the lateral hypothalamus includes orexin neurons, which stimulate nutrition, and melanin-concentrating hormone neurons, which inhibit hunger. These neurons subsequently project to the nuclei of the brainstem, where they are integrated with the peripheral input from the gastrointestinal system [95]. Furthermore, the pathological activation of the hypothalamus during migraine attacks can cause hyperphagia and weight gain. Interestingly, Zelissen et al. (1991) found higher levels of CGRP, a neuropeptide involved in the pathophysiology of migraine, in obese patients [96].

Given the potential impact that body weight can have on migraine outcomes, special attention should be paid to children’s and adolescents’ body weight and lifestyle. Emerging areas of therapeutic potential include dietary and lifestyle modifications, increased physical activity, and psychotherapy [67,73,79]. Interventions addressed to modify BMI may be relevant to reduce migraine severity. Psychological evaluation and psychotherapy may prevent a vicious cycle between weight, anxiety, and migraine [79].

## 5. Ketogenic Diet for Pediatric Migraine

The Ketogenic Diet (KD) is a safe and well-tolerated therapeutic tool for different metabolic and neurological disorders, including epilepsy, even during childhood. KD is a dietary regimen characterized by a severe depletion of carbohydrate intake, with a relative increase in fat consumption, leading metabolism to obtain energy from lipids through fatty acid oxidation and the formation of ketone bodies (KB). It has been used with success for more than a century for epileptic seizures, but its potential therapeutic efficacy on mi-graine has been poorly explored, above all in pediatric age. Different experimental models show that KD is able to reduce the propagation of cortical spreading depression [8], which plays a key role in the pathophysiology of migraine aura, and to decrease brain excitability by favoring GABAergic transmission [97]. Furthermore, KD and KB can inhibit neuroinflammation, oxidative stress, and free radical formation, which are processes involved in the pathophysiology of migraines [97,98,99,100,101,102]. In adults, KD can be an effective option, above all in patients requiring a hypocaloric diet for weight loss [99,100,101]. Different case reports have been published, suggesting the potential efficacy of KD, even in childhood [8,103,104,105]. In one case, KD proved to be effective in reducing hemiplegic migraine secondary to GLUT1 deficiency syndrome, with a recurrence of attacks in the case of a reduction of KB [106]. As for prospective studies, there is only one paper reporting an unsuccessful treatment of a group of 8 adolescents (13–16 years), but it should be underlined that they were treated with a modified Atkins Diet, which is a less strict diet that does not follow the same rigorous rules of the classic KD [107]. At present, the potential effect of a classic KD administered in children and adolescents suffering from chronic migraine has not been demonstrated.

## 6. Nutraceuticals—Therapeutic Nutrients

The term, nutraceutical, comes from the union between “nutrition” and “pharmaceutical”, and it could be described as “food, or parts of food, that provide medicinal or health benefits, including the prevention and treatment of disease” [108]. While the active ingredients of nutraceuticals are also present in common foods, it is necessary to resort to synthetic products in order to gain a therapeutic concentration of the active molecule.

The use of nutraceuticals for the treatment of migraine prophylaxis at pediatric age finds its strength in the tolerability of these molecules and the presence of low risks of side effects, compared to traditional pharmacological therapies [108]. Furthermore, the use of nutraceuticals may have been reinforced by a recent randomized study, showing that the efficacy of some traditional drugs was not different from that of a placebo [109]. On the other hand, the use of nutraceuticals at pediatric age has weak support from controlled studies versus a placebo [108,110]. 

Furthermore, while nutraceuticals are safe for the most part, it is wrong to think that the use of nutraceuticals is absolutely free of side effects as in the case of Petasites hybridus [111,112]. Finally, in the common imagination, the impression that they are natural substances often leads to self-medication, with the risk of under-treatment of the migraine or the conviction that therapeutic quantities of the active ingredient can be gained through one’s diet.

The most widely used nutraceuticals for the treatment of migraine are magnesium, riboflavin, coenzyme Q10, and polyunsaturated fatty acids. More recently, palmitoylethanolamide has also been used in the prophylaxis of migraine [113]. 

Magnesium is one of the most studied molecules for the treatment of migraine. It plays an important role in neuronal activity. Magnesium deficiency has been described in migraine patients, and this could support its therapeutic role in this disease [114]. A double-blind, randomized, controlled trial on 118 children evaluated the efficacy of oral magnesium (9 mg/kg/day for 12 to 16 weeks) for the prophylactic treatment of migraine. 

While a reduction in the frequency and intensity was observed in the magnesium group, this was not superior to a placebo. Regarding side effects, patients treated with magnesium were more likely to experience diarrhea or soft stools than those in the placebo group [115].

Coenzyme Q10 acts as an energy buster at the mitochondrial level. Its use is justified by the evidence of a deficit in mitochondrial functioning in migraine patients. A randomized controlled trial compared the efficacy of coenzyme Q10 (100 mg/day for 4 months) in reducing the frequency of the attacks of migraine in 62 pediatric patients. In this trial, both the CoQ10 group and the placebo group observed a reduction in the intensity and frequency of attacks, without statistical significance. No significant adverse events were reported [116].

The use of polyunsaturated fatty acids (PUFAs) is based on their anti-inflammatory effect. The most frequently used PUFAs are eicosapentaenoic (EPA), docosahexaenoic acid (DHA), and tocopherol. In a small randomized controlled trial conducted on 27 adolescents with chronic migraine, a fish oil compound containing EPA, DHA, and tocopherol showed equal efficacy as a placebo [117]. Another small double-blind trial randomized 25 children with migraine to either sodium valproate 20 mg/kg daily treatment with a fish oil compound containing EPA and DHA or to sodium valproate 20 mg/kg daily treatment with a placebo compound for 2 months. Both groups had a significant reduction in headache frequency and Pediatric Migraine Disability Assessment Scale (PedMIDAS) scores, but there were no statistically significant differences between the groups [118]. 

Riboflavin is a vitamin from the group of B vitamins (B2). It has a key role in cellular energy production during the oxidation–reduction reactions that occur in the mitochondria. Riboflavin is considered a valid option for the management of migraine in adults. However, non-adult studies appear to provide no such evidence [118,119]. Two retrospective case series involving children and adolescents with different types of headaches used riboflavin (200 mg–400 mg/day), without any evidence of efficacy for decision making in clinical practice [118,119]. 

Palmitoylethanolamide (PEA) is an endogenous fatty acid amide that is widely distributed in different tissues, including the nervous tissues. Its compound is naturally produced in many plant and animal food sources, as well as in the cells and tissues of mammals, and it is endowed with important neuroprotective, anti-inflammatory, and analgesic actions [120,121]. The anti-inflammatory effects of PEA seem to be mainly related to its ability to modulate mast cell (MC) activation and degranulation. MCs resident in the meninges can play a critical role in the development of inflammation in many diseases, including migraine [122,123]. To date, the efficacy of PEA for pediatric migraine prophylaxis has only been verified in an open label study. While this study has shown encouraging preliminary results, a comparison with a placebo group is necessary to gain clear clinical indications [124].

Limited data are available for the prophylaxis of migraine in children with melatonin [125], pyridoxine, vitamin B12, folate [126], and vitamin D [127].

In conclusion, the retrieved studies provide no clear evidence of the efficacy of nutraceuticals for the treatment of pediatric migraine. While nutraceuticals show some beneficial effects and an excellent tolerability in pediatric patients, additional research could potentially inform decision making. At this point, together with the evidence of the role of a placebo in pediatric migraine, we suggest the use of nutraceuticals in selected patients, such as very young children, where the side effects of traditional drugs are particularly alarming.

## 7. Conclusions

The literature data provide suggestions as to the role of nutrition in pediatric migraine. Regarding the role of food as a trigger factor, while some nutrients seem to play a role in inducing migraine attacks, it is necessary to verify the strict cause and effect relationship before making restrictive dietary changes. Some data suggest that a certain dietetic style that can prevent overweight and obesity is associated with a favorable evolution of migraine. Lastly, some nutrients can have a therapeutic role. The ketogenic diet offers encouraging results in adults, although the data are not available for children yet. While solid evidence of their efficacy in pediatric migraine is lacking, nutraceuticals are molecules that show a good tolerability profile, which allow them to be considered for use in younger patients or where there is a refusal to use traditional drugs for fear of the side effects. 

In conclusion, popular beliefs about the role of nutrients in pediatric migraine need to be substantiated by scientific studies, as science is the only tool that can allow us to consider such roles as “true”. If scientific evidence is lacking or even contradicts popular beliefs, that which is uncritically transmitted, e.g., the role of chocolate as a trigger factor of migraine, should be declassified as a “myth”.

## Figures and Tables

**Table 1 nutrients-13-02714-t001:** Main “truths” and “myths” in the migraine–nutrient relationship.

Topic	Truth	Myth
Can some foods trigger a migraine attack?	Some patients may have migraine attacks triggered by foods. Not all pro-migraine foods will trigger a headache attack in every patient with migraine. The most frequent ingredients with a pro-migraine effect are caffeine, monosodium glutamate, artificial sweeteners, nitrites, gluten, and biogenic amines. Eliminating some foods without a clear suspicion does not always improve migraines.	Migraine patients should eliminate trigger foods (e.g., chocolate and coffee) from their diet completely.
Can migraine be a symptom of food allergy?	Lack of studies. Migraine and food allergy are two common conditions that can coexist in the same child.	The Ig E-mediated allergic mechanism is involved in the pathogenesis of migraine. Migraine alone could be a symptom of food allergy. All migraine patients should be tested for food allergies (cutaneous prick test or IgE dosage).
Is there a relationship between migraines and obesity?	Obese children have more severe phenotypes of migraine (intense attacks and high frequency). Obesity can impact the severity of migraine by favoring the psychological comorbidities associated with it.	Obesity is a clear risk factor for migraine development. A decrease in body weight always leads to a reduction of migraine severity.
Ketogenic diet for migraine	The ketogenic diet counteracts phenomena implicated in the pathogenesis of migraine (cortical spreading depression and impaired GABAergic transmission). Ketogenic diet can be an effective option in adult patients, mostly in those requiring a hypocaloric diet for weight loss.	
Nutraceuticals	There are few controlled data on the efficacy of nutraceuticals in the treatment of pediatric migraine. Nutraceuticals are generally well-tolerated therapies.	The intake of either synthetic nutraceuticals or foods containing the active ingredients are equivalent. Nutraceuticals can be self-managed by patients and parents, without medical supervision. Nutraceuticals are always safe and free of side effects.

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
