# Peer review of "Truths and Myths in Pediatric Migraine and Nutrition"

_nutrients, 2021, doi:10.3390/nu13082714_

Round 1
Reviewer 1 Report
The current review brings to the surface several assumption and hypothesis regarding nutrition and pediatric migraine. My comments are :
- Throughout the article and especially in the " Foods as trigger factors in pediatric migraine" section, some of the studies that you refer to are based on an adult population. The focus of this review is supposed to be pediatric population. In my opinion , it should be clearly stated which information concern adults and maybe group them together in the text.
- Line 39: patients with overweight: change to overweight patients
- Line 133: reffering to red wine
- "Ketogenic diet for pediatric migraine": You mention that ketogenic diet has been shown to reduce cortical spreading depression. It would be interesting to include studies on the effect of ketogenic diet on hemiplegic migraine.
Author Response
Reviewer 1
Foods as trigger factors in pediatric migraine" section.
we have specified whether the studies were conducted on adult populations.
Ketogenic diet.
thank you for this interesting tip. A short mention of KD use in HM has been included (page 7, lines 302-303)
Reviewer 2 Report
The present article addresses an important and interesting issue in paediatric migraine. Moderate to extensive (in certain parts) English editing is required by a native speaker. Also, please correct the numbering of sections.
- Foods as trigger factors in pediatric migraine
When discussing the evidence for each individual food, please report the study design from which this evidence was derived. How certain are we about our findings? A well designed RCT is enough to inform clinical decision making. A well designed A mal-designed review on the other hand could provide false evidence. It is important that you critically appraise the quality of evidence and provide the reader with the necessary information to do so him/herself.
For example, regarding chocolate you mention that ‘all provocative studies have failed to confirm that chocolate can trigger migraine attacks.’. Please explain the design of a provocative study (blinded? Random consumption of chocolate or placebo?).
Also, with respect to caffeine and alcohol, are there any non-adult studies investigated their effect on migraineurs? Does all evidence stem from adult studies?
Lines 157-160: Please provide a possible explanation for the heterogeneity of the reported results. Based on your reporting, the reader cannot reach any conclusion regarding the triggering nature of aspartame. Similarly, with regards to MSG, please explain ‘weak evidence’ and discuss more the design and results of provocative studies. Finally, apart from the reported case report (Ref 60), what other evidence is there for the triggering nature of nitrites?
- Migraine and obesity.
I suspect that you misuse the definition of ‘clinical trial’ (line 263). Please check the corresponding line and confirm the correct – mistaken use of ‘clinical trial’.
- Ketogenic diet for pediatric migraine.
You mention that ‘Different anecdotal case reports have been published’, but anecdotal means unpublished.
You dedicate a part of this section to the interplay between obesity, physical activity and mood disorders. Is it clear which among these is the main determinant of migraine occurrence (or burden)? More importantly, is obesity a strong determinant, or its effect vanishes after adjusting for these confounders (mediating factor)?
- Nutraceuticals – Therapeutic nutrients
Lines 343-349: Please state explicitly that according to the results of the mentioned RCT (Ref 121) magnesium’ s efficacy proved equal to placebo, while its tolerability was inferior to placebo. Reporting within group differences could confuse the reader who might think that magnesium actually proved efficacious.
Lines 352-356: Similar to magnesium, please explicitly state between group differences. Please avoid using words such as ‘‘trend’’ because they usually confuse the reader and manipulate the translation of non-significant results. To my understanding the efficacy and tolerability of Co-Q10 was similar to placebo. If so, please clearly state that.
Lines 360-368: Alike before, please clearly state that fish oil was equally efficient to placebo.
Lines 360-368: Riboflavin is considered a valid option for the management of migraine in adults. However, non-adult studies appear to provide no such evidence. A retrospective (uncontrolled) case series (involving patients with headache in general) is as good as no evidence for decision making in clinical practice (Ref 125). Please summarize that there is no evidence for the potential efficacy of riboflavin, as above.
Please treat Ref 131 as above (open label uncontrolled studies cannot be taken into consideration in decision making).
‘In conclusion, there is little evidence on the efficacy of nutraceuticals for the treatment of 395 pediatric migraine. However, there seems to be a beneficial role for nutraceuticals.’: Please reformulate your conclusion. Retrieved studies provide no evidence (contrary to little). Therefore, additional research could potentially inform decision making (if you believe that addressing pitfalls of previous studies may modify their findings), but according to your article no evidence exists, to date.
Please refer to the current review series of Liampas et al. regarding the evidence for the use of several nutrients in migraine. Evidence was gathered and separately analysed for both adults and children. Vitamin D (Vitamin D serum levels in patients with migraine: A meta-analysis. Rev Neurol (Paris). 2020), pyridoxine, B12, folate (Pyridoxine, folate and cobalamin for migraine: A systematic review. Acta Neurol Scand. 2020 & Serum Homocysteine, Pyridoxine, Folate, and Vitamin B12 Levels in Migraine: Systematic Review and Meta-Analysis. Headache. 2020.) and melatonin (Endogenous Melatonin Levels and Therapeutic Use of Exogenous Melatonin in Migraine: Systematic Review and Meta-Analysis. Headache. 2020) are discussed. Please report the findings of these studies regarding the non-adult population.
Table 1: Please correct the word ‘Mith’ in the 3rd column.
Author Response
Reviewer 2
Foods as trigger factors in pediatric migraine
- We reported the study desing as requested (RCT, review etc); regarding chocolate we explained the design of the study; regarding caffeine and alcohol we specified when studies reffered to adults or children.
- Lines 157-160: we reported that There were fewer studies of nitrites triggering headaches in recent years.
Ketogenic diet
Answer: thank you, we deleted “anecdotal”.
Nutraceuticals – Therapeutic nutrients
Lines 343-349: We have modified the sentence.
Lines 352-356: We have modified the sentence.
Lines 360-368: We have modified the sentence.
Lines 360-368 : We summarized that there is no evidence for the potential efficacy of riboflavin.
Ref 131: we have modified the paragraph
We reformulated your conclusion.
We have added a reference to the suggested studies (melatonin, vitamin D, riboflavin etc).
In Table 1we corrected the word ‘Mith’ in the 3rd column.
Migraine and obesity
Thank you for the advice. We changed “clinical trial” into “retrospective study” (line 255)
Nutraceuticals – Therapeutic nutrients
Lines 343-349: We stated explicitly that magnesium’ s efficacy proved equal to placebo.
Lines 352-356: We stated explicitly that Co-Q10’ s efficacy proved equal to placebo.
Lines 360-368: We stated explicitly that fish oil was equally efficient to placebo.
Lines 360-368 and ref 131: We summarized that there is no evidence for the potential efficacy of riboflavin in children.
We reformulated our conclusion as suggested.
We added the suggested references.
Table 1: We corrects the word ‘Mith’ in the 3rd column.
English is being corrected by your english editing service
Round 2
Reviewer 2 Report
I have no further comments